# Therapeutic Cancer Vaccines in Colorectal Cancer: Platforms, Mechanisms, and Combinations

**DOI:** 10.3390/cancers17152582

**Published:** 2025-08-06

**Authors:** Chiara Gallio, Luca Esposito, Alessandro Passardi

**Affiliations:** Department of Medical Oncology, IRCCS Istituto Romagnolo per lo Studio dei Tumori (IRST) “Dino Amadori”, Via P. Maroncelli 40, 47014 Meldola, Italy; chiara.gallio@irst.emr.it (C.G.); alessandro.passardi@irst.emr.it (A.P.)

**Keywords:** colorectal cancer, cancer vaccines, immunotherapy, tumor cell-based vaccine, dendritic cell vaccine, peptide vaccine, mRNA vaccine, DNA vaccine, viral vector, tumor microenvironment

## Abstract

Colorectal cancer (CRC) remains a major global health concern, especially due to its high rates of recurrence and metastasis. In recent years, researchers have been developing cancer vaccines that train the immune system to recognize and attack cancer cells more effectively. This review explains how different types of cancer vaccines are being tested for CRC, including those made from tumor cells, dendritic cells, peptides, nucleic acids, and viruses. It also discusses how these vaccines may work better when combined with other treatments such as immunotherapy or chemotherapy. Although several challenges still exist, such as tumor variability and immune resistance, the progress made so far highlights the promise of vaccines as future options for preventing or treating CRC.

## 1. Introduction

Colorectal cancer (CRC) ranks as the third most prevalent malignancy in the world and represents the second leading cause of cancer-related mortality [1]. Despite advances in surgery, chemotherapy, and targeted therapies, recurrence and metastasis remain significant obstacles [2]. Immunotherapy, which leverages the body’s immune system to combat malignant cells, has emerged as a great promise in cancer treatment [3]. However, in CRC, its efficacy is primarily restricted to immune checkpoint inhibitors (ICIs) in a minority subset of patients—approximately 15% who present microsatellite instability-high (MSI-H)/mismatch repair deficiency (dMMR) cancers [4]. Recent clinical evidences enhance the role of combination immunotherapy in this subgroup of patients; in the CheckMate 8HW trial, a combination of monoclonal antibodies targeting programmed death-ligand 1 (PD-L1), i.e., Nivolumab, and cytotoxic T lymphocyte-associated antigen-4 (CTLA-4), i.e., Ipilimumab, significantly improved outcomes in the metastatic MSI-H/dMMR CRC subpopulation compared to Nivolumab alone [5]. Additionally, preliminary data from neoadjuvant immunotherapy trials in locally advanced dMMR CRC showed a promising efficacy in the early setting of cancer care [6,7]. In stark contrast, proficient mismatch repair (pMMR) CRC showed minimal clinical benefit from immunotherapy or chemoimmunotherapy regimens across several clinical trials [8,9,10]. The underlying mechanism of resistance remains incompletely understood in this subpopulation. Nevertheless, a crucial role is attributed to the presence of an immunosuppressive tumor microenvironment (TME) and to a low tumor mutational burden (TMB) [11]. In fact, the paucity of mutation-associated neoantigens impairs the development a proinflammatory TME, consisting of a high concentration of tumor-infiltrating lymphocytes (TILs), memory cells, and cytotoxic T lymphocytes (CTLs) [12]. Consequently, an efficient strategy to enrich proinflammatory mediators and potentiate anti-tumor immune response in CRC remains an unmet need.

In this context, cancer vaccines represent a highly promising approach due to their potential to stimulate long-lasting, tumor-specific immune responses. By presenting tumor-associated antigens (TAAs) or tumor-specific antigens (TSAs), cancer vaccines activate immune cells such as cytotoxic T lymphocytes (CTLs), contributing to sustained immune surveillance and potentially reducing recurrence and progression rates [13].

This review summarizes the latest advancements in CRC vaccine research, highlighting the different vaccine types under investigation, clinical trial outcomes, challenges in development, and potential future directions for vaccine-based immunotherapy. A brief flow chart outlining the methodology of literature selection is presented in Figure 1. The different mechanisms of action of CRC vaccines analyzed in this review are shown in Figure 2.

## 2. Cell-Based Vaccines in Colorectal Cancer Therapy

### 2.1. Tumor Cell-Based Vaccines: Enhancing Tumor Immunogenicity

Tumor cell-based vaccines use modified autologous cancer cells to stimulate immune responses against tumor cells (Table 1). A notable example is the Vigil™ vaccine, which was tested in a Phase I trial involving advanced solid tumors [14]. This autologous vaccine is engineered with granulocyte–macrophage colony-stimulating factor (GM-CSF) and a short hairpin RNA (shRNA) construct that enhances immune activation. In combination with FOLFOX-6 chemotherapy, Vigil™ showed prolonged disease-free survival in two patients, highlighting its clinical potential [15].

Another example is OncoVAX^®^, which involves irradiated autologous tumor cells combined with Bacillus Calmette–Guérin (BCG) as an immune adjuvant. Clinical trials of OncoVAX^®^ demonstrated the feasibility of stimulating a patient’s immune response against their own colorectal tumor [16]. The ECOG 5283 Phase III trial confirmed immunogenicity, though without showing disease-free survival (DFS) benefit [17]. The subsequent 8701 Phase III trial added BCG and a six-month booster inoculation after the initial three weekly treatments in Stage II and III CRC. According to stratification analysis by disease stage, OncoVAX^®^ significantly improved 5-year overall survival (OS) compared to surgery alone in Stage II patients (5-year OS 82.5% versus 72.7% in the control group, *p* value = 0.014) [18].

Due to the emergence of FOLFOX as standard care, Stage III enrollment was halted, though combination protocols are now FDA-approved. A bioequivalent sterile version of OncoVAX^®^ has been developed to meet FDA requirements, preserving efficacy and enabling large-scale manufacturing [19].

A Phase 2 study evaluated the combination of the GVAX colon vaccine—a tumor cell-based vaccine derived from irradiated GM-CSF-secreting allogeneic colon cancer cells—with low-dose cyclophosphamide and pembrolizumab in 17 patients with advanced pMMR CRC. The combination was well tolerated; however, only 3 of 17 patients achieved a disease control rate (DCR) at their first radiological assessment. Although the primary endpoint was not met, the addition of GVAX to pembrolizumab induced a significant decline in carcinoembryonic antigen (CEA) levels, an effect not observed in the prior clinical trials of pembrolizumab monotherapy for pMMR CRC. These results suggest that this combination, including cyclophosphamide, may enhance the immune response to PD-1 inhibition [20].

### 2.2. Dendritic Cell-Based Vaccines: Directing the Immune Response

Dendritic cell-based vaccines involve isolating a patient’s dendritic cells, loading them with tumor antigens, and reinfusing them to prime immune responses. Dendritic cells (DCs) include a wide repertoire of antigen-presenting cells (APCs), such as conventional DCs, Langerhans cells, plasmacytoid DCs, and monocyte-derived DCs [21]. Several clinical trials have shown that these vaccines can effectively enhance immune responses and, in some cases, improve clinical outcome in CRC patients.

For example, a Phase II trial involving autologous tumor lysate-pulsed DCs in refractory metastatic CRC increased cytokine production but did not improve survival outcomes compared to the best supportive care alone. Despite the induction of tumor-specific T cells, the absence of clinical benefits highlighted the need of immune-modulating therapy combination to improve efficacy [22]. Moreover, a Phase I/II trial at Duke Cancer Institute studied a CEA RNA-pulsed DC vaccine on 24 patients with a high risk of recurrence after the resection of metastatic CEA-expressing malignancy, the majority of whom had colon cancer. The vaccine was well tolerated and capable of inducing tumor-specific immune responses [23].

The GEMCAD 1602 Phase I-II study evaluated the safety and efficacy of avelumab combined with autologous dendritic cell vaccine in microsatellite stable (MSS) metastatic CRC patients. The combination was safe and showed modest clinical activity with evidence of enhanced immune responses [24]. Another Phase I/II randomized study evaluated DC vaccines with or without CD40 ligand (CD40L) activation in CRC patients. CD40L activation enhanced tumor-specific immune responses, and immune-responders reported improved RFS at 5 years compared to non-responders (63% vs. 18%, *p* value = 0.037), suggesting a potential immunologic benefit [25].

### 2.3. Induce Pluripotent Stem Cells (iPSCs): Increasing Response by Expressing Tumor-Associated Antigens

Emerging research has investigated the potential of using induced pluripotent steam cells (iPSCs) as cancer vaccines, leveraging their ability to express TAAs. iPSCs share key similarities with tumor cells, including pluripotency and self-renewal capacity, and are employed in modeling disease pathogenesis [26,27]. In the context of cancer immunotherapy, the intrinsic heterogeneity of iPSCs enabled the expression of a wide repertoire of epitopes, which has been shown to elicit a robust prophylactic immune response [28]. Comparative studies between iPSCs and human embryonic stem cells (hESCs) underlined the similarity in immuno-responsiveness across different applications [29]. However, autologous iPSCs are less likely to trigger alloimmune responses than hESCs and overcome ethical concerns associated with hESC use [30].

Preclinical murine studies, conducted on melanoma, breast cancer, and mesothelioma, have demonstrated that irradiated iPSC-based vaccines prevented tumor growth and induced a tumor-specific T cell immune response [31]. Recently, a group of upregulated genes, referred as iPSC-cancer signature genes, was identified through the cluster analysis of RNA sequencing data from iPSCs and various cancer cell lines. Vaccination with iPSCs enriched for this signature resulted in a marked increase in CD8+ T cell infiltration within tumor sites. Conversely, the depletion of T cells abolished the anti-tumor activity, highlighting their essential role in driving the immune response. In particular, HNRNPU and NCL have been recognized as key targets that elicit strong immune responses. The benefit was evidenced both in primary and in metastatic CRC mouse models. This approach could enable universal PSC transplantation, through the generation of hypoimmunogenic autologous iPSC lines, eliminating the need to create an autologous iPSC line for each individual patient [32].

**Table 1 cancers-17-02582-t001:** Cell-based cancer vaccines in colorectal cancer.

Vaccine	Type of Study (N of pts Enrolled)	Clinical Stage at Enrollment	Adjuvant	Combination	Primary Endpoint(s)	Result	Reference
**Tumor Cell-Based Vaccines**							
Vigil™	Case report (2)	Stage IV colon cancer (liver metastases)	GM-CSF	FOLFOX-6	Safety, immune response	DFS > 8 years for both patients; positive immuno-responsiveness	[15]
OncoVAX^®^	Phase III (254)	Stage II and III colorectal cancer after surgery	BCG	/	RFS	Vaccination showed 41% risk reduction for disease progression (5-y survival, *p* = 0.008) Major impact on Stage II disease	[18]
GVAX	Phase II (17)	Late-stage Stage IV pMMR colorectal cancer	GM-CSF	Cyclophosphamide + pembrolizumab	ORR	No ORR observed CEA decline in 41% of pts	[20]
**Dendritic Cell-Based Vaccines**							
Autologous tumor lysate-pulsed DC vaccine	Phase II (52)	Late-stage metastatic colorectal cancer	/	/	4-month PFS	Median PFS 2.7 vs. 2.3 mo (*p* = 0.628) Tumor-specific T cell responders had longer OS (7.3 vs. 3.8 mo)	[22]
CEA RNA-pulsed DC vaccine	Phase I/II (13 in Phase II analysis)	Metastatic CEA-expressing tumors (including CRC) undergone radical surgery	/	/	Safety; feasibility	No grade II-IV toxicity reported Generation of cytotoxic CEA-specific T cells	[23]
Autologous tumor lysate DC vaccine (GEMCAD 1602)	Phase I/II (19)	Late-stage MSS metastatic CRC	/	Avelumab	20% 6-month PFS increase	Six-month PFS 11%; median PFS 3.1 mo (2.1–5.3 mo); median OS 12.2 mo (3.2–23.2 mo)	[24]
CD40L-activated autologous DC vaccine	Phase II (26)	Stage II and III colorectal cancer after surgery	CD40L (12 of 26 pts)	/	Tumor-specific immune response	Tumor-specific immune response rate 61% Improved 5-year RFS in responders (63% vs. 18%, *p* = 0.037)	[25]
**Induced Pluripotent Stem Cell (iPSC)-Based Vaccines**							
Autologous iPSC-based vaccine	Preclinical	Animal models	CpG	/	Tumor-specific immune response	Induced T cell responses inhibited tumor growth	[32]

**BCG**: Bacillus Calmette–Guérin; **CEA**: carcinoembryonic antigen; **CRC**: colorectal cancer; **CpG**: Cytosine-phosphate-Guanine; **DC**: dendritic cell; **DFS**: disease-free survival; **FOLFOX**-**6**: 5-fluorouracil, leucovorin, and oxaliplatin; **GM-CSF**: granulocyte–macrophage colony-stimulating factor; **iPSC**: induced pluripotent stem cell; **MSS**: microsatellite stable; **ORR**: objective response rate; **OS**: overall survival; **PFS**: progression-free survival; **pMMR**: proficient mismatch repair; **pts**: patients; **RFS**: recurrence-free survival.

## 3. Peptide-Based Vaccines in Colorectal Cancer Treatment

Peptide-based cancer vaccines aim to activate immune responses against cancer cells using synthetic peptides that target specific tumor antigens. Several clinical trials have investigated peptide-based cancer vaccines in CRC patients (Table 2). In a Phase 1 study, Taniguchi et al. investigated OCV-C02, a peptide-based vaccine targeting two colorectal cancer-associated epitopes, in patients with refractory metastatic colorectal cancer. The vaccine was well tolerated, with no dose-limiting toxicities observed. Immunogenicity analysis revealed peptide-specific T cell responses in a portion of treated patients [33].

In the Phase 1 AMPLIFY-201 trial, Pant et al. evaluated a novel lymph-node-targeted amphiphile vaccine designed to elicit T cell responses against mutant KRAS (mKRAS) in patients with advanced pancreatic and colorectal cancers. The vaccine, administered in combination with anti-PD-1 therapy, demonstrated an acceptable safety profile and induced mKRAS-specific CD8+ T cell responses in a subset of patients [34].

In a double-blind, placebo-controlled Phase II trial, a MUC1 peptide vaccine was tested for the prevention of recurrent colorectal adenomas in high-risk individuals. The vaccine was immunogenic and showed a non-significant trend toward reduced adenoma recurrence, suggesting potential for colorectal cancer prevention pending further validation [35].

A Phase I/II study evaluated the immunological impact of adding interferon-α to the p53-SLP^®^ vaccine, a synthetic long peptide targeting p53 mutations, in colorectal cancer patients. The addition of interferon-α significantly enhanced IFN-γ production by vaccine-induced T cells, reflecting a more potent Th1 immune response. The findings support the potential of interferon-α as an immune adjuvant to improve vaccine efficacy in p53-expressing colorectal tumors [36].

The Phase 2 trial Immatics IMA910 evaluated IMA910, a multi-peptide vaccine targeting 13 tumor-associated antigens, in HLA-A*02-positive patients with advanced CRC and stable disease post-chemotherapy. Vaccinations with IMA910 plus granulocyte-macrophage colony-stimulating factor (GM-CSF), with or without imiquimod, following low-dose cyclophosphamide, were well tolerated. Patients mounting T cell responses to ≥2 peptides showed better disease control and longer progression-free and overall survival. Median OS exceeded 28 months in multi-T cell responders versus 16 months in non-responders. A matched analysis comparing the intention-to-treat population of the study with COIN trial patients also suggested a survival benefit (median OS 19.7 vs. 16.5 months, Hazard Ratio = 0.665; *p* value = 0.0386) [37].

In another trial, peptides derived from TOMM34, RNF43, and VEGFR significantly increased IgG levels against these antigens; notably, anti-VEGFR IgG levels correlated with improved overall survival in advanced CRC patients [38].

A series of Japanese clinical trials evaluated a seven-peptide cancer vaccine targeting multiple tumor-associated antigens in combination with oral chemotherapy (tegafur-uracil plus leucovorin) in patients with advanced or metastatic CRC. Across several Phase I and I/II studies, the combination regimen demonstrated a favorable safety profile and induced peptide-specific cytotoxic T lymphocyte (CTL) responses in a substantial proportion of patients. Clinical outcomes included stable disease and potential survival benefit in select cases, suggesting a synergistic effect between vaccine-induced immune responses and oral chemotherapy [39].

HER2 has also emerged as a vaccine target: in a study by Hattori et al., metastatic CRC patients received personalized peptide vaccines (targeting SART2/3, MDR-associated protein 3, HER2/neu, cytochrome B, ubiquitin-conjugating enzyme E2, and CEA) in combination with standard 5-fluorouracil-based chemotherapy, showing clinical benefit and strong immune responses [40]. A Phase I trial (NCT01376505) testing a HER2/neu peptide vaccine with nor-MDP adjuvant in Montanide has completed the dose-escalation phase. An extension cohort enrolled patients with advanced solid tumors, including CRC, with no serious adverse events or dose-limiting toxicities reported to date [41].

**Table 2 cancers-17-02582-t002:** Peptide-based cancer vaccines in colorectal cancer.

Vaccine	Type of Study (N of pts Enrolled)	Clinical Stage at Enrollment	Adjuvant	Combination	Primary Endpoint(s)	Result	Reference
OCV-C02 peptide vaccine	Phase I (24)	Stage IV colon cancer	Montanide ISA 51 VG	/	Safety	No dose-limiting toxicities Induced T cell responses	[33]
mKRAS G12D/G12R amphiphile peptide (ELI-002 2P)	Phase I	Stage IV KRAS-mut CRC/PDAC with MRD after resection	Amph-CpG-7909	/	Safety Recommended Phase 2 dose	No dose-limiting toxicities mKRAS-specific T cell response rate 84% PFS improvement in responders	[34]
MUC1 peptide	Phase I/II (103, 53 vaccines, 50 placebo)	Colorectal adenoma	Polyinosinic-polycytidylic acid	/	Immune response at 12 wks	No significant reduction in adenoma recurrence Immuno-responders at weeks 12 showed 38% absolute reduction compared to placebo (66%)	[35]
p53-SLP^®^	Phase I/II (11)	Stage IV colon cancer in remission		Interferon-alpha	Safety; immunogenicity	No relevant toxicities Marked increase in IFN-γ T cells	[36]
IMA910 multi-peptide	Phase I/II (92)	Stage IV colon cancer post oxaliplatin-based therapy in stable disease	GM-CSF±topical imiquimod	Cyclophosphamide pre-vaccine (immunomodulation)	Immunogenicity; OS vs. matched COIN arm C	Median OS 19.7 mo vs. 16.5 mo (HR 0.675, *p* = 0.047) Immuno-responders reported improved OS than non-responders	[37]
Multi-peptide personalized vaccine	Phase II (89)	Stage IV colon cancer	GM-CSF	Oxaliplatin-based chemotherapy	Safety; immunogenicity; IgG/CTL biomarker correlation with OS	Elevated IgG to TOMM34, RNF43, VEGFR2; VEGFR2 IgG linked with improved OS; CTL response increased, but not correlated with OS	[38]
Seven-peptide cocktail vaccine	Phase I/II (30)	Stage IV colon cancer	Montanide ISA 51 VG	Tegafur-uracil plus leucovorin	Safety; Feasibility	One case of anaphylaxis and no other serious adverse events were reported Positive CTL responders had longer OS	[39]
Poly-peptide personalized vaccine	Phase I (14)	Stage IV colon cancer	Montanide ISA 51	Tegafur-uracil plus leucovorin	Safety; immunogenicity	Good tolerance IgG response correlated with increased OS (*p* = 0.0215) CTL response increased, but not correlated with OS or PFS	[40]
HER2 chimeric B cell peptide vaccines	Phase I (49)	Late-Stage IV solid tumors (including CRC)	Montanide ISA 720VG + nor-MDP	/	Safety; immunogenicity	No relevant toxicities IgG response Poor disease control rate	[41]

**CRC**: colorectal cancer; **CTL**: cytotoxic T lymphocyte; **GM-CSF**: granulocyte–macrophage colony-stimulating factor; **HER2**: Human Epidermal Growth Factor Receptor 2; **HR**: Hazard Ratio; **IFN** γ: interferon gamma; **IgG**: Immunoglobulin G; **MRD**: minimal residual disease; **mo**: months; **OS**: overall survival; **PFS**: progression-free survival; **pts**: patients; **VEGFR2**: Vascular Endothelial Growth Factor Receptor 2; **wks**: weeks.

## 4. Nucleic Acid-Based Vaccines

### 4.1. mRNA Vaccines

mRNA vaccines have garnered attention for their ability to induce both humoral and cellular immune responses. Several clinical trials have explored their potential in CRC treatment (Table 3). A Phase I clinical trial evaluated mRNA-5671/V941 vaccine in metastatic malignant solid tumors as monotherapy and in combination with pembrolizumab; the vaccine targets four of the most commonly occurring KRAS mutations (G12D, G12V, G13D, and G12C) [42]. Despite preliminary positive results, the trial has been prematurely closed due to business reasons. Other mRNA-based vaccines, including BNT122, have demonstrated effectiveness in combination with immune checkpoint inhibitors, and future data will come from a dedicated trial on adjuvant CRC therapy (NCT04486378) [43].

### 4.2. DNA Vaccines

DNA vaccines utilize bacterial plasmids encoding TAAs, triggering antigen-specific immune response. Gribben et al. conducted a Phase I clinical trial evaluating the ZYC300 vaccine in patients with advanced, pre-treated solid tumors. The vaccine, which encoded the CYP1B1 antigen, was delivered via biodegradable microparticles. Among the 17 patients treated, no significant adverse events were reported. Remarkably, five patients who exhibited immune responses to CYP1B1 showed marked and durable responses to their subsequent line of therapies. These findings highlight the potential role of cancer vaccines in combination strategies aimed at enhancing their clinical efficacy [44]. The MYPHISMO study, presented by Pham et al., outlines a phase I trial combining the TetMYB cancer vaccine with anti-PD-1 checkpoint blockade in patients with advanced solid tumors, including CRC and adenoid cystic carcinoma. The trial explores a novel immune modulatory strategy targeting MYB, a transcription factor implicated in oncogenesis. Although results are pending, the study represents an innovative approach to personalized immunotherapy through transcription factor-targeted vaccination in combination with PD-1 inhibition [45]. Preclinical studies also suggest that multi-epitope neoantigen DNA vaccines could enhance immune responses and induce effective anti-tumor immunity [46].

**Table 3 cancers-17-02582-t003:** Nucleic acid-based cancer vaccines in colorectal cancer.

Vaccine	Type of Study (N of pts Enrolled)	Clinical Stage at Enrollment	Adjuvant	Combination	Primary Endpoint(s)	Result	Reference
mRNA-5671/V941	Phase I/II (16 CRC pts)	Metastatic KRAS-mut (G12D, G12V, G13D or G12C) solid tumors (including CRC)	/	±Pembrolizumab	Safety	Closed prematurely for business reasons	[42]
BNT122 (Autogene cevumeran)	Phase I/II (213, 12 CRC pts)	Late-Stage IV solid tumors (including CRC)	/	±Atezolizumab	Safety; tolerability	Mild-to-moderate systemic reactions Long-lasting neoantigen-specific T cell response rate was similar in both monotherapy and combination therapy	[43]
ZYC300	Phase I (17, 3 CRC pts)	Late-Stage IV solid tumors (including CRC)	/	/	Safety; immunogenicity	Well-tolerated; immune response to CYP1B1 induced in one of three CRC patients Immuno-responders had better response to subsequent therapy	[44]
TetMYB	Phase I (27)	Late-Stage IV solid tumors (including CRC)	/	+BGB-A317 (anti PD-1)	Safety; immunogenicity	No results published yet	[45]

**CRC**: colorectal cancer; **CYP1B1**: Cytochrome P450 Family 1 Subfamily B Member 1; **KRAS**: Kirsten Rat Sarcoma Viral Oncogene Homolog; **PD-1**: programmed death-1; **pts**: patients.

## 5. Virus-Based Vaccines

Virus-based vaccines leverage viral vectors to deliver tumor antigens, stimulating immune responses. Adenoviral vaccines have been tested in CRC patients and have shown the potential to induce strong immune responses (Table 4). Crosby et al. evaluated VRP-CEA (6D), an alphavirus vector, in 28 patients with Stage IV CRC. Due to slow accrual, an expansion cohort of 12 additional Stage III patients was enrolled. The treatment enhanced the CD8+ effector memory T cell-to-regulatory T cell (Treg) ratio and improved long-term survival in Stage III CRC patients, who exhibited a 5-year RFS rate of 75% (95% confidence interval (CI) = 40–91%) with no reported deaths. These findings suggest that vaccination administered after surgical resection and adjuvant chemotherapy may enhance clinical outcomes, offering a greater promise than in heavily pre-treated metastatic disease [47].

The Ad5-GUCY2C-PADRE vaccine, targeting guanylate cyclase 2C (GUCY2C) overexpressed in CRC cells, has demonstrated durable T cell responses in early-phase trials. However, the presence of pre-existing anti-Ad5 neutralizing antibodies seemed to impair vaccine-induced immune responses, limiting its efficacy [48]. In this context, a Phase IIA clinical trial is currently testing a chimeric viral vector composed of the Ad5 capsid and Ad35 fiber protein (Ad5.F35), aiming to improve these promising results [49].

Oncolytic virus-based vaccines, such as those using adenoviruses, can also reshape the tumor microenvironment and enhance the immune response against CRC. Balint et al. presented extended results from a Phase I/II trial evaluating the Ad5 [E1-, E2b-]-CEA(6D) vaccine, an advanced-generation adenoviral vector encoding a modified CEA, in patients with late-stage CRC. The vaccine demonstrated sustained immunogenicity and a favorable safety profile, with no severe adverse events reported. Notably, patients who received an optimal vaccination schedule reported a modest improvement in overall survival (29-month OS = 23%). These data suggest that enhanced adenoviral vectors targeting tumor-associated antigens like CEA may provide therapeutic benefit in advanced disease settings [50].

A randomized Phase II trial tested AdCEA vaccine and avelumab added to mFOLFOX6 and bevacizumab as the first-line treatment of metastatic colorectal cancer. Although the immunotherapy combination was safe, it did not significantly improve progression-free survival compared to chemotherapy alone [51]. Gögenur et al. conducted a Phase 1/2 trial assessing the immunologic and clinical effects of neoadjuvant intratumoral influenza vaccine administration in patients with pMMR CRC. The treatment resulted in increased the intratumoral infiltration of CD8+ T cells and the upregulation of PD-L1 expression, suggesting a potential for synergy with immune checkpoint inhibitors [52].

**Table 4 cancers-17-02582-t004:** Virus-based cancer vaccines in colorectal cancer.

Vaccine	Type of Study (N of pts Enrolled)	Clinical Stage at Enrollment	Adjuvant	Combination	Primary Endpoint(s)	Result	Reference
VRP-CEA (6D)	Phase I/II (40)	Stage III and IV CRC	/	/	Safety, immunogenicity, CD8+ T cell response, overall survival	Well tolerated; increased CD8+ effector memory T cell-to-Treg ratio and long-term survival benefit in Stage III colon cancer patients (5-year RFS = 75%)	[47]
Ad5.F35-GUCY2C-PADRE	Phase I/II (43 of 81)	High-risk gastrointestinal adenocarcinomas after radical surgery	/	/	Safety, immunogenicity	Ongoing trial, no results published yet	[49]
Ad5 [E1-, E2b-]-CEA(6D)	Phase I/II (32)	Late-stage IV CRC	/	/	Safety, immunogenicity, overall survival	Well tolerated; increased CEA-specific T cell responses; patients who received optimal vaccination schedule reported higher OS	[50]
AdCEA	Phase II	Untreated pMMR metastatic CRC	/	mFOLFOX6 + bevacizumab + avelumab	PFS	No significant differences in mPFS: 8.8 months (95% CI: 3.3–17.0 mo) versus 10.1 months (95% CI: 3.6–16.1 mo) in control arm Increased CD4+ and CD8+ T cell responses	[51]
Intratumoral influenza vaccine	Phase I/II (10)	Stage II-III pMMR CRC patients before curative surgery	/	/	Safety	No adverse events recorded; increased intratumoral CD8^+^ T cells; enhanced PD-L1 protein expression	[52]

**Ad5**: Adenovirus Serotype 5; **AdCEA**: adenovirus-based CEA vaccine; **CEA**: carcinoembryonic antigen; **CI**: confidence interval; **CRC**: colorectal cancer; **GUCY2C**: Guanylyl Cyclase C; **mFOLFOX6**: modified FOLFOX-6 (5-FU, leucovorin, oxaliplatin); **mo**: months; **mPFS**: median progression-free survival; **OS**: overall survival; **PD-L1**: programmed death-ligand 1; **pMMR**: proficient mismatch repair; **PFS**: progression-free survival; **pts**: patients; **RFS**: recurrence-free survival; **Treg**: regulatory T cells.

## 6. Challenges in Cancer Vaccine Development

Despite the substantial advances in CRC vaccine research, several critical challenges continue to hinder the widespread clinical application of these therapies. One of the foremost obstacles is tumor heterogeneity. CRC exhibits extensive inter- and intra-patient variability at the genetic, epigenetic, and antigenic levels, which limits the efficacy of “one-size-fits-all” vaccines and highlights the need for personalized or precision vaccine strategies [53]. Designing vaccines that effectively target tumor-specific neoantigens, while accounting for the constantly evolving mutational landscape, remains a key area of focus [54].

Another major issue is the presence of immune evasion mechanisms employed by CRC tumors. These include the downregulation of antigen presentation pathways, secretion of immunosuppressive cytokines, upregulation of immune checkpoint molecules (e.g., PD-L1), and recruitment of Tregs and myeloid-derived suppressor cells (MDSCs) within the TME. These adaptations contribute to a highly immunosuppressive milieu that can blunt vaccine-induced immune responses [55,56].

The limited immunogenicity of tumor-associated antigens also represents a considerable barrier. Many self-derived tumor antigens fail to elicit robust immune responses due to immune tolerance or insufficient T cell activation [57]. Consequently, the use of potent adjuvants, delivery platforms, or combination regimens with immune checkpoint inhibitors, chemotherapy, or radiation therapy is often necessary to boost immunogenicity and achieve therapeutic benefit [51,56,58].

Moreover, the complexity of vaccine design, delivery, and manufacturing—particularly for autologous or neoantigen-based vaccines—adds significant cost and logistical burdens. These processes require high-throughput sequencing, bioinformatic analyses, and Good Manufacturing Practice (GMP)-compliant production of patient-specific formulations, often within tight timelines [59]. A recent publication by Novakova et al. highlighted the possible pricing scenarios of peptide-base vaccines for triple-negative breast cancers. A lack of efficacy for the standalone administration of cancer vaccines needs to focus on maximizing the life prolongation factor and minimizing the number of doses per patient in order to enhance sustainability [60]. Manufacturing innovation and AI-guided design implementation have shortened the production timelines for personalized vaccines in recent years. Despite these improvements, production costs remain prohibitive mainly due to manufacturing costs. Regulatory frameworks have evolved to support accelerated development pathways in order to improve research activity in cancer vaccine development. However, the scalability and accessibility of such personalized approaches are expected to remain limited especially in low-resource settings [61].

## 7. Future Directions

Recent advances in genome sequencing and computational biology offer promising opportunities for innovative strategies for cancer vaccines. Current research is focused both on the identification of immunogenic neoantigens as potential vaccine targets and on refining vaccine design through state-of-the-art bioengineering approaches.

An untiring effort is dedicated to the identification of novel epitopes, focusing on those capable of durable and localized immune responses. The widespread adoption of high-throughput DNA and RNA sequencing technologies permitted to create a comprehensive bioinformatic database useful for emulating cancer vaccine response to relevant neoantigens [62]. Although in silico analyses are not able to provide data on the durability of response or the pharmacokinetics of cancer vaccines, they offer valuable support in optimizing cost-effectiveness and reducing development timelines. By minimizing the reliance on animal models and potentially substituting early-phase trials, in silico modeling contributes to significant resource savings and streamlines the preclinical evaluation process [63,64].

Among in silico approaches, reverse vaccinology represents a promising strategy. The principle of this innovative methodology relies on eliminating the necessity of cultivating pathogens, leveraging genomic data to identify potential vaccine targets. Furthermore, it is especially advantageous for pathogens that are difficult or impossible to culture under laboratory conditions [65]. Recently, the development of mRNA-based vaccines against severe acute respiratory syndrome coronavirus 2 (SARS-CoV-2) has generated enhanced interest in the broader application of this approach across other areas of medicine, including oncology [66].

Reverse vaccinology application in cancer vaccine design is limited and is largely confined to pre-clinical trials [67,68]. In CRC, this approach has been explored in preventive strategies targeting microbiota related to cancer developments. Despite promising potential in terms of immunogenicity and theoretical efficacy, its effectiveness in cancer prevention has not been adequately validated and requires further investigation [69,70].

Nanovaccines represent another considerable advancement in vaccine design. Recent progress in nanotechnology has enhanced the delivery of TAAs to APCs, ameliorating both immune-responsiveness and the biodistribution of nanoparticle-based platforms [71]. Nanovaccines are classified into four main classes: polymeric, lipid, inorganic, and biologically derived types. Several trials are investigating their applicability in cancer care programs [72,73].

Another emerging axis of innovation lies in the microbiome–immunity interface. The gut microbiota has been shown to modulate host immunity and influence both tumor progression and clinical outcomes [74]. These findings have led to the exploration of microbiome-based adjuvants and microbiota-targeted vaccines, particularly for prophylactic interventions in high-risk individuals [75,76,77].

Such strategies could play a pivotal role in cancer prevention, adding a new dimension to the utility of vaccines beyond treatment. Recent publications about therapeutic vaccines in colorectal cancer are summarized in Table 5.

## 8. Expert Opinion

The landscape of CRC immunotherapy is rapidly evolving, and cancer vaccines are emerging as one of its most promising components. While they are still far from being the standard of care, recent research has reignited interest in their therapeutic and preventive potential. Instead of directly confronting the immunosuppressive TME of bulky solid tumors, therapeutic vaccines may prove most valuable when deployed against minimal residual disease (MRD), circulating tumor cells (CTCs), or early micro recurrences—where the immune system has a better chance of regaining control [34,83].

Despite encouraging progress, several challenges continue to retard the integration of cancer vaccines into clinical practice. Most reported trials showed modest benefits on clinical outcomes, highlighting the compelling necessity to implement vaccination with comprehensive cancer care. The development of highly effective cancer vaccines depends on the identification of tumor antigens with robust immunogenicity, strong binding affinity to human leukocyte antigen (HLA), and precise delivery technologies [84,85]. Platforms such as lipid nanoparticles, viral vectors, and DC carriers are currently being optimized to ensure antigen retention in lymphoid tissues and minimize off-target immune responses. These innovations are critical for enhancing vaccine efficacy while reducing toxicity [86].

Another relevant strategy to overcome a lack of efficacy lies on exploring a combination of cancer vaccines with immuno-enhancing therapies.

In this context, cancer vaccines are increasingly being investigated in a combination with immune checkpoint inhibitors (ICIs), adoptive T cell therapies, cytokine-based treatments, and oncolytic viruses. Unlike conventional prophylactic vaccines, therapeutic cancer vaccines are designed to prime T cell engagement to the TME. However, cancer vaccine monotherapy has generally reported limited efficacy due to reduced longevity and suboptimal activity at tumor sites [56]. In order to overcome these limitations, combining ICIs with cancer vaccines could amplify pre-existing T cell-mediated immune responses, initially enhanced by cancer vaccine [87,88]. In addition, conventional chemotherapy implementation in a combination strategy showed immune responses within the TME, promoting macrophage polarization and antigen presentation to APCs [89]. These combination strategies aim to intensify immune activation, sustain cytotoxic T cell function, and overcome immune tolerance in MSS CRC—traditionally resistant to immunotherapy [90].

Personalized vaccine design is further supported by the identification of predictive biomarkers. For example, the altered expression of specific circulating microRNAs—such as miR-6826, miR-6875, miR-125b-1, or miR-378a—has been correlated with poor vaccine response and reduced OS in CRC patients [91,92]. Several studies on advanced CRC showed how to identify patients who can benefit from immunotherapy.

Polymerase epsilon and delta (POLE/POLD1) mutations have been proposed as biomarkers identifying a subtype of MSS CRC highly responsive to ICIs [93]. Similarly, the dMMR status has been extensively validated as a prognostic and predictive factor of mCRC [94]. Emerging data from neoadjuvant immunotherapy trials in dMMR locally advanced CRC also indicate promising clinical outcomes, enhancing immune-modulating regimens at an early stage [6,7].

As for immunotherapy, a high TMB is also being investigated as a predictive biomarker for mRNA-based vaccine therapy [95].

Recently, Liu et al. used bioinformatics analysis on CRC transcriptome data from The Cancer Genome Atlas (TCGA) to identify four immune subtypes, based on over 1900 immune-related gene expression. Survival analysis revealed that the elevated expression of *THBS2*, *FSTL3*, *TNNT1*, *BGN*, *CTHRC1*, and *NOX4* was associated with poor OS in CRC. Notably, these genes were especially enriched in immune subtypes 2 and 4, which also exhibited a higher TMB, suggesting that they may serve as promising candidates for mRNA vaccine development [96].

However, specific predictive biomarkers evaluating CRC vaccine responsiveness need to be evaluated. Incorporating such biomarkers into clinical trial designs could guide patient selection and improve response rates.

This article presents a comprehensive and timely synthesis of current vaccine-based strategies in CRC, exploring well-known approaches such as autologous tumor cell vaccines, dendritic cell-based formulations, peptide epitope targeting, nucleic acid platforms, and emerging studies on iPSC-based vaccines, nanovaccines, and reverse vaccinology. A notable strength lies in the balanced integration of preclinical data with early-phase clinical trial outcomes, including vaccine modalities that have reached Phase III evaluation (e.g., IMA910, CEA-targeted viral constructs). A novel emphasis will come from ongoing clinical trials exploring vaccine combination with checkpoint inhibitors and nanodelivery systems to overcome the immunosuppressive tumor microenvironment typical of pMMR CRC. An improvement in vaccine design including in silico modeling and AI-guided neoantigen identifications highlight the awareness of evolving precision immunotherapy paradigms.

However, several limitations need to be pointed out. Since the article analyzes a wide range of vaccine approaches, deeper insights into the mentioned articles are needed to enhance the complexity and the heterogeneity of the vaccine framework. Potential biases may arise from the selective inclusion of positive data, while heterogeneity in study design across trials limits direct comparisons. Moreover, many of the described strategies remain at preclinical or early clinical stages, and the discussion of translational barriers—such as manufacturing scalability, immunogenicity durability, and cost-effectiveness—is limited. These aspects will require focused investigation to inform future clinical application.

## 9. Conclusions

Vaccine strategies tailored to individual tumor profiles, immune phenotypes, and microbial signatures represent a powerful vision for the future of oncology. While it is still premature to consider CRC vaccines as mainstream therapy, ongoing studies and multidisciplinary innovation are laying the groundwork for their integration into personalized treatment protocols. Ultimately, the long-term goal remains ambitious yet tangible: to develop cancer vaccines capable not only of treating but also of preventing tumor formation through a single, well-targeted immunization. Continued translational research and clinical validation will determine whether this goal can be achieved, but the current trajectory suggests that CRC vaccines may one day become a cornerstone of comprehensive cancer care.

## Figures and Tables

**Figure 1 cancers-17-02582-f001:**
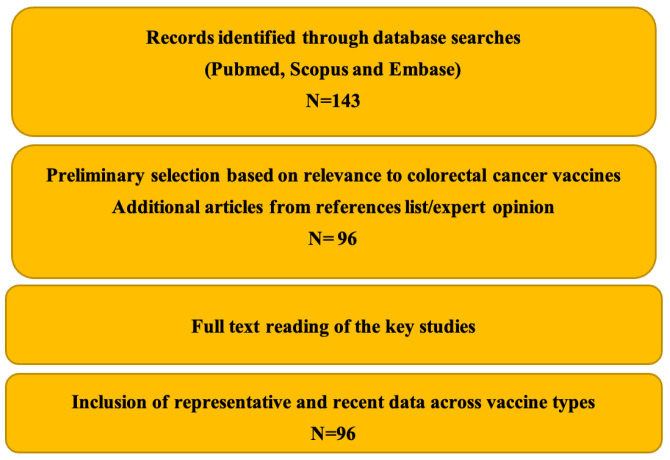
Flow chart of literature selection.

**Figure 2 cancers-17-02582-f002:**
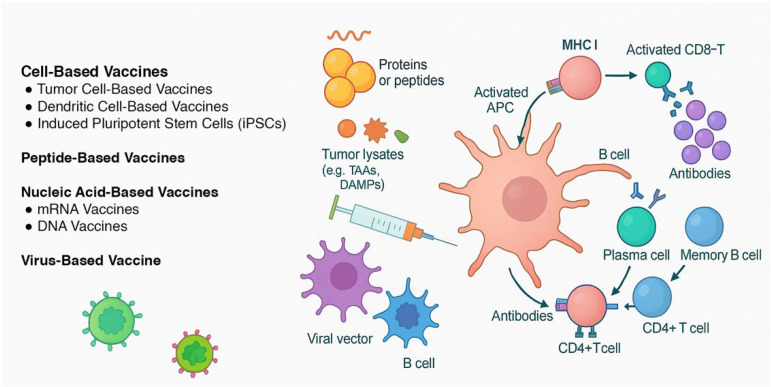
The different mechanisms of action of vaccines developed for colorectal cancer. Cell-based vaccines: Tumor cell-based vaccines use autologous or allogeneic whole cancer cells—modified (e.g., GM-CSF, BCG, shRNA) or irradiated—to present a broad array of tumor-associated antigens (TAAs) and stimulate a polyclonal immune response. Dendritic cell-based vaccines are designed by patient-derived dendritic cells, which are pulsed with tumor lysates or antigen-encoding RNA (e.g., CEA) and reinfused to direct antigen presentation and T cell priming. Enhancements like CD40L activation or checkpoint blockade (e.g., avelumab) aim to increase efficacy. iPSC-based vaccines exploit the tumor-like antigenic profile of induced pluripotent stem cells, which express a wide TAA repertoire. Irradiated iPSCs elicit CD8^+^ T cell–mediated responses and have shown anti-tumor activity in preclinical CRC models via cancer-specific gene signatures. Peptide-based vaccines use synthetic peptides derived from tumor antigens, which are presented by APCs to stimulate adaptive immunity. Nucleic acid-based vaccines (DNA/mRNA) encode genetic material from tumor antigens, leading to endogenous expression and T cell activation. Virus-based vaccines modulate viral vectors that express tumor antigens or directly lyse cancer cells, triggering strong immune responses.

**Table 5 cancers-17-02582-t005:** Recent clinical trials of therapeutic vaccines in colorectal cancer.

Vaccine	Trial ID	Type of Study	Clinical Stage at Enrollment	Route	Adjuvant	Combination	Primary Endpoint(s)	Secondary Endpoints	Key Findings	Reference
PolyPEPI1018	NCT03391232 (OBERTO-101)	Phase IIb	MSS metastatic CRC	Subcutaneous (arms and thighs)	Montanide ISA51VG	Fluoropyrimidine/ bevacizumab maintenance	Safety	Efficacy, immunogenicity at peripheral and tumor level, immune correlates	Well tolerated; enhanced multigenic CD8+ T cell response; improved PFS in longer vaccination schedule	[78]
GV1001	/	Phase II	Recurrent or refractory mCRC	Intradermal	/	Chemotherapy	DCR	ORR PFS, OS, safety	DCR was 90–9% No detectable vaccine-specific immune response; no improvement in OS and PFS	[79]
Personalized neoantigen peptide vaccine	ChiCTR1900022372 (China)	Phase II	Recurrence of metastatic MSS CRC	Subcutaneous (axillary and groin areas)	Polyinosinic:polycytidylic acid (Poly I:C)	/	Safety; feasibility	Immunogenicity, PFS	Well tolerated; T cell responses in 4 of 6 patients PFS improved in responders (19 vs. 11 months)	[80]
ChAd68 mRNA vaccine	NCT03639714	Phase I	Advanced solid tumors (including MSS CRC)	Intramuscular	/	Nivolumab+Ipilimumab	Safety Tolerability Recommended Phase 2 dose	Feasibility, OS, immunogenicity, ctDNA clearance	No dose-limiting toxicity; CD8+ T cell activation observed No notable improvement in survival outcomes	[81]
NeoAg-VAX	/	Phase I	Metastatic MSS CRC	Subcutaneous	Imiquimod	±Pembrolizumab	Safety, feasibility	ORR, PFS, 12-week PFS rate, T cell response	No dose-limiting toxicity; enhanced neoantigen-specific T cell responses; positive correlation between T cell responders and intratumor immune cell density at baseline	[82]

**ctDNA**: circulating tumor DNA; **CRC**: colorectal cancer; **DCR**: disease control rate; **MSS**: microsatellite stable; **mCRC**: metastatic colorectal cancer; **ORR**: objective response rate; **OS**: overall survival; **PFS**: progression-free survival; **Poly I:C**: polyinosinic:polycytidylic acid (synthetic analog of double-stranded RNA); **pts**: patients; **T cell**: T lymphocyte.

## Data Availability

No new data were created or analyzed in this study. Data sharing is not applicable to this article.

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
