# Peer review of "Therapeutic Cancer Vaccines in Colorectal Cancer: Platforms, Mechanisms, and Combinations"

_cancers, 2025, doi:10.3390/cancers17152582_

Round 1
Reviewer 1 Report
Comments and Suggestions for Authors
This review article submitted to Cancers J by MDPI, titled “Therapeutic Cancer Vaccines in Colorectal Cancer: Platforms, Mechanisms, and Combinations” by Gallio et al., 2025
- In the current research the authors addressed the peptide’s safety profile in repeated dosing, combined with evidence of no systemic accumulation, supporting the development of new regimens involving higher and more frequent doses to enhance antitumor efficacy in clinical studies.
- The topic is original and relevant to the field and interesting
The abstract is better to be structured, with details,- Key words are enough,
- In the Introduction, the first paragraph is without ref. and this is not correct “
- Figure 1 should be moved forward after the mechanisms mentioned in the figure,
- The aim is well written
- Again, long sentences without references.
- Table 1 findings are they according to the clinical trials.gov?
- What about applying reversed vaccinology approach?
- The personalized vaccine in line 264 lacks references
- List of abbreviations is provided
Minor corrections:
- The introduction is obscure and needs clarification and more details,
- Several sentences are without ref.
- Please split long sentences and add a ref. for each single sentence or each single info.
- Figure 1 needs to be reallocated
- The “strength(s)” of the study to be mentioned
- Limitations to be added
- Add the future directions
- The challenges’ part lacks references as well as the expert opinion part
Major corrections:
- add a flowchart of the work design,
- Need one table to enumerate the cell-based, peptide-based, and finally NA-based vaccines and so on
- Need several tables summarizing findings in each part.
- Each vaccine-based part needs a separate figure and table to summarize and enumerate findings
- Add reversed vaccinology approach
- No conclusion is there
- References by authors are few
Author Response
Dear reviewer,
thank you so much for your spending your time having a look on our review. In order to explain clearly what we have improved you will find an exhaustive description of the modifications in the uploaded file below.
Please see the attachment.
Kindly regards,
Dr. Luca Esposito

Reviewer 2 Report
Comments and Suggestions for Authors
Dear Authors,
I have reviewed your manuscript "Therapeutic Cancer Vaccines in Colorectal Cancer: Platforms, Mechanisms, and Combinations" and find it to be a comprehensive and well-structured review that addresses an important and timely topic in cancer immunotherapy. The manuscript provides valuable insights into the current state of colorectal cancer vaccine development across multiple platforms and would be of significant interest to the readership.
Your systematic coverage of all major vaccine platforms (cell-based, peptide-based, nucleic acid-based, and virus-based) is commendable, and the inclusion of recent clinical trial data enhances the clinical relevance of the work. The expert opinion section provides thoughtful perspectives on future directions and challenges in the field.
However, several areas require attention to strengthen the manuscript's impact and clarity. The review would benefit from more critical analysis of the clinical data presented, particularly regarding the modest benefits observed in many trials. Additionally, the discussion of predictive biomarkers and patient selection criteria deserves expansion given the heterogeneity challenges inherent in colorectal cancer.
I recommend acceptance of this manuscript after addressing the following minor revisions:
Minor comments:
- Please provide more critical analysis of the clinical trial outcomes presented throughout the manuscript. Many trials show modest clinical benefits - this should be discussed more explicitly with potential explanations for limited efficacy and strategies to overcome these limitations.
- The biomarker discussion (lines 264-268) is too brief given its importance for patient selection. Please expand this section to include more details about predictive biomarkers, their validation status, and potential clinical applications in vaccine selection.
- Line 98: Please correct "Induce Pluripotent Steam Cells" to "Induced Pluripotent Stem Cells" and ensure the mechanism by which iPSCs function as cancer vaccines is more clearly explained.
- Figure 1 could be improved with clearer labeling and color-coding to better distinguish between different vaccine types and their mechanisms of action. Consider adding a legend that better explains the various cellular interactions depicted.
- Table 1 contains abbreviations that are not defined in the text (e.g., "PolyPEPI1018"). Please ensure all abbreviations are either defined within the table or added to the comprehensive abbreviation list.
- The combination strategies discussion is scattered throughout the manuscript. Consider adding a dedicated subsection within the challenges/future directions section that specifically addresses combination approaches (vaccines + checkpoint inhibitors, vaccines + chemotherapy) given their promising nature.
- Please include confidence intervals where available when mentioning survival benefits (e.g., the "20% survival advantage" mentioned on line 68).
- The cost-effectiveness considerations mentioned in the challenges section deserve brief elaboration, particularly given the complexity and personalized nature of many vaccine approaches discussed.
Author Response
Dear reviewer,
thank you so much for your spending your time having a look on our review.
Please see the attachment to find an exhaustive description of what we have done to resolve the issues you pointed out.
We also provide to update a refined version of the manuscript.
Kindly regards,
Dr. Luca Esposito

Round 2
Reviewer 1 Report
Comments and Suggestions for Authors
All comments done